# Investigation on Mg_3_Sb_2_/Mg_2_Si Heterogeneous Nucleation Interface Using Density Functional Theory

**DOI:** 10.3390/ma13071681

**Published:** 2020-04-03

**Authors:** Mingjie Wang, Guowei Zhang, Hong Xu, Yizheng Fu

**Affiliations:** School of Materials Science and Engineering, North University of China, Shanxi 030051, China; 15513882577@163.com (M.W.); hhm727@163.com (G.Z.); wmj313999@126.com (Y.F.)

**Keywords:** first-principles method, Heterogeneous nucleation, interfacial energy, Mg_2_Si (111)/Mg_3_Sb_2_ (0001) interface

## Abstract

In this study, the cohesive energy, interfacial energy, electronic structure, and bonding of Mg_2_Si (111)/Mg_3_Sb_2_ (0001) were investigated by using the first-principles method based on density functional theory. Meanwhile, the mechanism of the Mg_3_Sb_2_ heterogeneous nucleation potency on Mg_2_Si grains was revealed. The results indicated that the Mg_3_Sb_2_ (0001) slab and the Mg_2_Si (111) slab achieved bulk-like characteristics when the atomic layers N ≥ 11, and the work of adhesion of the hollow-site (HCP) stacking structure (the interfacial Sb atom located on top of the Si atom in the second layer of Mg_2_Si) was larger than that of the other stacking structures. For the four HCP stacking structures, the Sb-terminated Mg_3_Sb_2_/Si-terminated Mg_2_Si interface with a hollow site showed the largest work of adhesion and the smallest interfacial energy, which implied the strongest stability among 12 different interface models. In addition, the difference in the charge density and the partial density of states indicated that the electronic structure of the Si-HCP-Sb interface presented a strong covalent, and the bonding of the Si-HCP-Mg interface and the Mg-HCP-Sb interface was a mixture of a covalent bond and a metallic bond, while the Mg-HCP-Mg interfacial bonding corresponded to metallicity. As a result, the Mg_2_Si was conducive to form a nucleus on the Sb-terminated-hollow-site Mg_3_Sb_2_ (0001) surface, and the Mg_3_Sb_2_ particles promoted the Mg_2_Si heterogeneous nucleation, which was consistent with the experimental expectations.

## 1. Introduction

Aluminium-magnesium-silicon alloys have shown considerable promise as the universal candidate materials for automotive and aerospace applications because of the formation of a Mg_2_Si heterogeneous nucleus [1,2]. Computational simulations and experimental reports have elucidated the potency of Mg_2_Si as a heterogeneous particle to reinforce α-Al and α-Mg nucleation in aluminium and magnesium alloys, respectively [3,4,5]. Although the Mg_2_Si compound has a high hardness and elastic modulus, the coarse primary Mg_2_Si phase, that has also been called Chinese script Mg_2_Si in the existing literature, that appears in the Mg-Si alloy cannot meet the requirements of engineering performance [6,7,8]. Thus, the heterogeneous nuclei for the refinement of the coarse Mg_2_Si phase are considered the most effective method to achieve specific engineering-designed requirements [9,10]. To date, the attention has been on the grain-refining efficiency, for the Mg_2_Si phase, of adding an alterant [11,12,13,14,15] to Mg-Si alloys. For instance, Ba_2_Sb, CaSb_2_, Mg_3_Sb_2_, Li_2_Sb, and Mg_3_P_2_ show a positive efficiency in the grain improvement of Mg_2_Si. Variations with various rare-earth elements, such as Y, La, Nd, and Gd [16,17,18,19], have been reported. Among the three common Sb-based master alloys, the lattice parameters of antimony trimagnesium have the lowest mismatch with the Mg_2_Si nucleation phase [20]. Therefore, it is understandable that Mg_3_Sb_2_ may refine the size of Mg_2_Si.

The theoretical derivation of the interfacial properties and the interrelationships of phases at the interface based on density functional theory (DFT) has been widely used to predicate heterogeneous nucleation [21]. According to the literature, the morphology of Mg_2_Si is inclined to transform into an octahedron shape from Chinese script Mg_2_Si through surface anisotropy by the first-principles simulation, when Sb is doped in a Mg_2_Si crystal [22]. Previous studies have proven that the formation of Mg_3_(Sb, Si)_2_ and Mg_3_Sb_2_ particles can act as the heterogeneous nuclei of primary Mg_2_Si to refine the particle size of the primary Mg_2_Si crystals [23]. However, the research on the structure and interfacial characteristics of a heterogeneous nucleation interface between the Mg_2_Si phase and the Mg_3_Sb_2_ substrates has been predominantly ignored.

Recent studies have been carried out using theoretical evidence to investigate the nucleation potential of a heterogeneous substrate through research on the properties of the bulk, interface stability, and interfacial energy of a heterogeneous nucleation interface [24,25,26]. A first-principles calculation with the density functional theory (DFT) as an atomic analysis method has been widely implemented to illustrate heterogeneous nucleation [27,28]. Theoretical estimates of heterogeneous nucleation between two solid interfaces have been mainly based on the Bramfitt mismatch theory [29], which elucidates that the smaller the mismatch of two heterogeneous lattice structures is, the smaller the interface energy is and the more effective the heterogeneous core growth is. 

In this work, the surface energy, work of adhesion, interfacial energy, and electronic properties of Mg_2_Si (111)/Mg_3_Sb_2_ (0001) interfaces were investigated through the density functional theory, which provides theoretical support for Mg_3_Sb_2_ as the heterogeneous nucleation substrates of Mg_2_Si grains, and which lays the theoretical foundation for the grain refinement of aluminium-magnesium alloys. Because the crystal structures of Mg_2_Si and Mg_3_Sb_2_ are cubic and trigonal lattices respectively, we chose to study the Mg_2_Si (111)/Mg_3_Sb_2_ (0001) interface. According to the equation of Bramfitt, the lattice mismatch of Mg_2_Si (111)/Mg_3_Sb_2_ (0001) is only 2.02%.

## 2. Computational Methodology

The interfacial and surface properties of Mg_2_Si (111)/Mg_3_Sb_2_ (0001), such as surface energies, work of adhesion, and interfacial bonding energies, were implemented in the Cambridge serial total energy package (CASTEP) code based on the density functional theory [30,31]. To calculate the self-consistent electronic density, the generalized gradient approximation (GGA) [32,33] with PW91 functional was performed to obtain the exchange-correction function in this study. The valence electrons of Mg, Si, and Sb, calculated in terms of their pseudopotentials, were 2p^6^3s^2^, 3s^2^3p^2^, and 5s^2^5p^3^4d^10^, respectively. All the plane wave cut-off energies for the bulk, surface, and interface were selected as 520 eV, the value of the k point was set as 10 × 10 × 10 for bulk Mg_2_Si and Mg_3_Sb_2_, and that for their surface and interface was set to 6 × 6 × 1 and 8 × 8 × 1, respectively. The self-consistent field (SCF) convergence threshold was set as 1.0 × 10^−6^ eV/atom to solve the Kohn-Sham equation, and the equilibrium crystal structure was obtained using the Broyden Fletcher Goldfrab and Shanno (BFGS) method. Moreover, the convergence tolerances for energy changes, force tolerance, stress, and displacement tolerance were set to 1.0 × 10^−6^ eV/atom, 0.03 eV/Å, 0.05 GPa, and 0.01 Å, respectively.

## 3. Bulk and Surface Properties

### 3.1. Bulk Properties of Mg_2_Si and Mg_3_Sb_2_

To estimate the reliability of the computational methods, space groups, the lattice constants, bulk modulus, and elastic constants for bulk Mg_2_Si and bulk Mg_3_Sb_2_, as listed in Table 1, were implemented using the density functional theory. From Table 1, the crystal structures of Mg_2_Si and Mg_3_Sb_2_, as shown in Figure 1, are cubic and trigonal crystal systems with the space groups of Fm$\bar{3}$m and P$\bar{3}$m1, respectively. The calculated results were in reasonable agreement with the previous theoretical calculations and experimental data [34,35,36,37], which verified the reliability of the calculations. Moreover, to obtain further insight into the bonding types of bulk Mg_2_Si and bulk Mg_3_Sb_2_, the total and partial densities of the states for bulk Mg_2_Si and bulk Mg_3_Sb_2_ were investigated, as shown in Figure 2. This figure clearly shows that the major conduction band states were Mg 2p orbitals for both bulk Mg_2_Si and bulk Mg_3_Sb_2_, which indicated that the metallic bonding existed in the Mg_2_Si and Mg_3_Sb_2_ phase. Moreover, Figure 2a shows that from −9.5 eV to −7.5 eV, the vast majority of the valence band states were Si 3s, and that from −4.5 eV to the Fermi level a considerable majority of the conduction band states were Si 3p, which suggested that the bonding in the Mg_2_Si phase included both metallic bonds and covalent bonds.

### 3.2. Surface Properties of Mg_2_Si(111) and Mg_3_Sb_2_(0001)

The convergence test for the different thickness slabs of Mg_2_Si (111) and Mg_3_Sb_2_ (0001) was significant for bulk-like interiors to ensure a sufficient thickness of the interface. Thus, the convergence tests of the Mg_2_Si (111) and Mg_3_Sb_2_ (0001) slabs were performed first to confirm whether the optimal number of layers was appropriate for the bulk-like interior. Commonly, the calculation accuracy of the obtained results increased with an increasing number of layers. Therefore, the selection of the number of layers, considering the cost of the computational time, applied to the convergence test.

The surface energy of the Mg-based phase variation with various terminated atoms was used as one of the important parameters to elucidate the surface stability. The calculation of surface energy can be expressed as follows [38]:(1)$$E_{surface}=\frac{1}{2A_{surface}}\left[E_{slab}\left(N\right)-NE_{bulk}\right]$$
where *E_slab_(N)* is the total energy of the slab, *E_bulk_* is the bulk energy per layer of the Mg-based bulk after optimisation, *A_surface_* is the surface area, and *N* is the number of surface slabs. Moreover, the odd-numbered slabs were selected to eliminate the influence of the polar surface on the computational results, and a vacuum gap (10 Å) was inserted on the surface of Mg_2_Si(111) and Mg_3_Sb_2_(0001) to erase the periodic effect between the surface atoms. 

To further determine the thickness of both the Mg_2_Si (111) slab and the Mg_3_Sb_2_ (0001) slab, different termination conditions were modelled, such as Mg-terminated and Si-terminated Mg_2_Si (111) slabs, and Mg-terminated and Sb-terminated Mg_3_Sb_2_ (0001) slabs. Different numbers of layers (5, 7, 9, and 11) were considered for the convergence tests of four different terminated slabs. Therefore, the distances between two adjacent layers after the surface relaxation of Mg_2_Si (111) and Mg_3_Sb_2_ (0001) with different terminations and slab thicknesses are presented in Table 2. This table shows that the interlayer relaxation change of both Mg_2_Si(111) and Mg_3_Sb_2_(0001) slabs exhibited a converging trend when the atomic layers were *N* ≥ 11. Therefore, 11-layer atoms of both Mg_2_Si(111) and Mg_3_Sb_2_(0001) slabs were constructed, and a four-surface model was built, as shown in Figure 3.

### 3.3. Stability of Mg_2_Si(111) and Mg_3_Sb_2_(0001) Surface 

The characteristics of the terminating atoms have a significant influence on the surface energy. Therefore, the surface energy of the Mg-terminated and Si-terminated Mg_2_Si (111) slabs and that of the Mg-terminated and Sb-terminated Mg_3_Sb_2_ (0001) slabs were investigated for further insight into the surface stability of the Mg_2_Si (111) and Mg_3_Sb_2_ (0001) surfaces. It was significant for the chemical potentials of different elements in the analysis of the phase transition and surface energy. Thus, the chemical potentials had to be considered in the calculation of the surface energy. The surface energy of the Mg_2_Si (111) plane could be expressed as follows [39,40]:(2)$$\sigma_{Mg_{2}Si}(111)=\frac{1}{2A}\left[E_{slab}-N_{Mg}\mu_{Mg}-N_{Si}\mu_{Si}+PV-TS\right]$$
where *E_slab_* is the total energy of a relaxed surface slab, *A* is the surface area of the surface structure, μ_i_ (i = Mg, Si) elucidates the chemical potentials of i atoms, and *N_Mg_* and *N_Si_* are the number of Mg and Si in the Mg_2_Si(111) slab, respectively. Due to the CASTEP being implemented under 0K and typical pressures, PV and TS could be ignored. In general, the surface slab was in equilibrium with the bulk structure after full relaxation; therefore, the chemical potentials of the Mg_2_Si(111) plane could be expressed by bulk Mg_2_Si as follows [41]:(3)$$\mu_{Mg_{2}Si}^{bulk}=2\mu_{Mg}+\mu_{Si}$$
(4)$$\mu_{Mg_{2}Si}^{bulk}=2\mu_{Mg}+\mu_{Si}+\Delta H$$
where $\mu_{\mathrm{Mg}}^{\mathrm{bulk}}$ and $\mu_{\mathrm{Si}}^{\mathrm{bulk}}$ are the total energy of the Mg and Si atoms in the pure metal Mg and Si, respectively; ΔH is the formation heat of bulk Mg_2_Si, which is calculated as follows:(5)$$\Delta H_{{\mathrm{Mg}}_{2}\mathrm{Si}}=(E_{total}-N_{Mg}E_{\mathrm{Mg}}-N_{Si}E_{Si})/(N_{Mg}+N_{Si})$$
where E_total_ is the total energy of a Mg_2_Si unit cell; N_Mg_ and N_Si_ are the number of Mg and Si atoms in a Mg_2_Si unit cell, respectively; and E_Mg_ and E_Si_ are the energies per Mg and Si atom, respectively. Considering the structural stability of the surface model, the chemical potentials should be lower and meet the following requirements: μ_Mg_ ≤ $\mu_{\mathrm{Mg}}^{\mathrm{bulk}}$ and μ_Si_ ≤ $\mu_{\mathrm{Si}}^{\mathrm{bulk}}$. Thus, by combining Equations (2) and (3), we calculated the range of Δμ_Mg_ = μ_Mg_ − $\mu_{\mathrm{Mg}}^{\mathrm{bulk}}$ and the surface energy as follows:(6)$$\frac{1}{2}\Delta H\leq \mu_{Mg}-\mu_{Mg}^{bulk}\leq 0$$
(7)$$\sigma_{Mg_{2}Si}\left(111\right)=\frac{1}{A}\left[E_{slab}-Ni_{Si}\mu_{Mg_{2}Si}^{bulk}+\left(2N_{Si}-N_{Mg}\right)\mu_{Mg}\right]$$

The surface energy of the Mg_3_Sb_2_ (0001) plane was also calculated by using the same method as that used for the Mg_2_Si (111) slab. Figure 4 shows the relationship between Δμ_M__g_ and the calculated surface energy of both Mg_2_Si (111) and Mg_3_Sb_2_ (0001) with different terminations. This figure shows that the surface energies of the Mg-terminated and the Si-terminated Mg_2_Si (111) were 1.425–1.546 J/m^2^ and 1.306–1.43 J/m^2^, respectively. Additionally, for the Mg_3_Sb_2_ (0001) slab, the formation heat was −2.135 eV; furthermore, the surface energies of the Mg-terminated and Sb-terminated Mg_3_Sb_2_ (0001) were 0.921–1.234 J/m^2^ and 1.023–1.479 J/m^2^, respectively. Moreover, the surface energy of the Mg_3_Sb_2_ (0001) slab and the Mg_2_Si (111) slab have to be higher than that of the α-Mg surface (0.58J/m^2^) from the viewpoint of reference [42]. These results showed that the surface energy of the Si-terminated Mg_2_Si (111) surface was smaller than that of the Mg-terminated Mg_2_Si (111) surface over the entire range, which indicated that the Si-terminated surface trended toward a stable value. Moreover, the surface energy of the Sb-terminated Mg_3_Sb_2_ (0001) surface was lower than that of the Mg-terminated Mg_3_Sb_2_ (0001) surface under the Sb-rich condition, but the Mg-terminated surface energy was lower under the Mg-rich condition.

## 4. Properties of the Mg_2_Si/Mg_3_Sb_2_ Interface

### 4.1. Mg_2_Si(111)/Mg_3_Sb_2_(0001) Interface Model

On the basis of the results of the convergence tests discussed above, the interface model of Mg_2_Si (111)/Mg_3_Sb_2_ (0001) was constructed with a superlattice geometry, which combined an 11-layer Mg_2_Si (111) slab and an 11-layer Mg_3_Sb_2_ (0001) slab. As both Mg_2_Si (111) and Mg_3_Sb_2_ (0001) had two different termination structures and three possible symmetry stacking sequences (OT, MT, and HCP), as shown in Figure 5, there were 12 possible Mg_2_Si (111)/Mg_3_Sb_2_ (0001) models, where the OT and the MT refer to the position of the bottom atom facing the top and the center of the first layer of another surface model, and the HCP refers to the position of the bottom atom facing the second layer of another surface model. Simultaneously, to reduce the number of interactions among the surface atoms, a vacuum layer of 15 Å was stacked on the substrate of the Mg_3_Sb_2_ (0001) surface. To keep the periodic boundary conditions, the coherent interface approximation was performed during the super-cell calculation [43,44].

### 4.2. Mg_2_Si(111)/Mg_3_Sb_2_(0001) Interface Stability

The work of adhesion (W_ad_), as a significant evaluation reference for the interfacial bonding strength, is the reversible work against the separation of interfacial atoms [45]. In general, a higher *W_ad_* represents a stronger binding ability of the interface, and the *W_ad_* of the Mg_2_Si/Mg_3_Sb_2_ interface can be expressed as follows:(8)$$W_{ad}=\left(E_{total}^{Mg_{2}Si}+E_{total}^{Mg_{3}Sb_{2}}-E_{total}^{Mg_{2}Si/Mg_{3}Sb_{2}}\right)/A$$
where $E_{\mathrm{total}}^{{\mathrm{Mg}}_{3}\mathrm{Si}}$ and $E_{\mathrm{total}}^{{\mathrm{Mg}}_{3}{\mathrm{Sb}}_{2}}$ are the total energy of the Mg_2_Si slab and the Mg_3_Sb_2_ slab after full relaxation, respectively; $E_{\mathrm{total}}^{{\mathrm{Mg}}_{2}\mathrm{Si}/{\mathrm{Mg}}_{3}{\mathrm{Sb}}_{2}}$ is the total energy of the Mg_2_Si/Mg_3_Sb_2_ interface; and A is the interface area.

In general, the energy of the Mg_2_Si slab and the Mg_3_Sb_2_ slab often remains the same for the one interface structure. Thus, the variation and the fitting curves of the unrelaxed interface energy and the interfacial distance (*d_0_*) were calculated first to obtain the optimal *W_ad_*, as shown in Figure 6. This figure shows that the total energy of 12 optimal interface models and *d_0_* of Mg-(OT, MT, HCP)-Sb, Mg-(OT, MT, HCP)-Mg, Si-(OT, MT, HCP)-Sb, and Si-(OT, MT, HCP)-Mg were obtained preliminarily. Moreover, it can be seen that the HCP interface exhibited a minimum interface energy and minimum interface distance compared with the other two stacking sequences (OT and MT). In contrast, the HCP stacking structure had a maximum *W_ad_*. Considering the computational cost and the acquirement of fully relaxed *W_ad_*, the calculation was performed without relaxation, and the interatomic interactions on distance were not considered. Therefore, the next step was to determine the revised optimal *W_ad_* and *d_0_* with different ‘optimal’ *d_0_* after full relaxation. 

The optimal *W_ad_* and *d_0_* results for the relaxed geometries of these 12 interfaces are listed in Table 3. Remarkably, a comparison of the *W_ad_* and *d_0_* of an unrelaxed interface with those of a fully relaxed interface revealed that *W_ad_* and *d_0_* exhibited a slight increase and decrease, respectively, after the full relaxation of the interface. This might be attributed to the interfacial charge redistribution and atomic displacement that occurred in the interface during the relaxation, resulting in considerable improvements in the bonding strength of the interface. In other words, the initial three stacking sequences of the interfacial structure were non-equilibrium states.

Table 3 shows that both Mg-terminated and Si-terminated Mg_2_Si(111) surfaces were likely to combine with the Sb-terminated Mg_3_Sb_2_(0001) surface, mainly because the interfacial bonding strength of the Mg–Mg and Si–Mg bonds was inferior to that of the Mg–Sb and Si–Sb bonds. Meanwhile, the interfacial bonding strength of the Si-terminated Mg_2_Si(111) surface combined with the Mg-terminated and Sb-terminated Mg_3_Sb_2_(0001) surface was stronger than that of the Mg-terminated Mg_2_Si(111) surface; this was possibly due to the higher bond strength of the covalent bond between Si and Mg, Sb than that of the metal bond between Mg and Mg, Sb, respectively.

Along with the ideal cohesive energy of the interface, the interfacial energy played a significant role in estimating the interfacial stability. The calculation formula of the interfacial energy of the Mg_2_Si/Mg_3_Sb_2_ interface can be expressed as follows [46,47]:(9)$$\gamma_{\mathrm{int}}=\sigma_{Mg_{2}Si}+\sigma_{Mg_{3}Sb_{2}}-W_{ad}$$
where $\sigma_{Mg_{2}Si}$ and $\sigma_{Mg_{3}Sb_{2}}$are the surface energy of the Mg_2_Si (111) and Mg_3_Sb_2_ (0001) slabs, respectively. *W*_*ad*_ is the work of adhesion of the Mg_2_Si/Mg_3_Sb_2_ interface.

Figure 7 compares the intercorrelations among the interfacial energies of 12 Mg_2_Si(111)/Mg_3_Sb_2_ (0001) interface models as a function of the Mg chemical potential. Compared with the OT and MT stacking structure, as shown in Figure 7, the interfacial energy of the HCP stacking structure was the lowest among all the terminations. In the whole scope of Δμ_Mg_, the interfacial energies for the Mg–HCP–Mg interface, Mg–HCP–Sb interface, Si–HCP–Mg interface, and Si–HCP–Sb interface were 1.486–1.726 J/m^2^, 1.215–1.515 J/m^2^, 0.18–0.42 J/m^2^, and 0.069–0.369 J/m^2^, respectively. This indicated a higher stability for the HCP stacking sequence of the Mg_2_Si/Mg_3_Sb_2_ interface. Moreover, the Si-terminated Mg_2_Si (111) and Sb-terminated Mg_3_Sb_2_ (0001) interface with the HCP stacking sequence had the lowest interfacial energy, which further implied that this interface configuration was the preferred equilibrium structure for the Mg_2_Si/Mg_3_Sb_2_ interface. Moreover, the Mg–HCP–Mg had the highest interfacial energy of all the HCP stacking structures, which indicated that it had a smaller interfacial stability than the other three interfaces. Simultaneously, all the results of the interfacial energy for the 12 models were well consistent with the results of *W*_*ad*_. Considering the efficiency and simplicity of the analysis, the next section mainly discusses the Mg–HCP–Mg interface, Mg–HCP–Sb interface, Si–HCP–Mg interface, and Si–HCP–Sb interface, because of the better interfacial stability of the HCP stacking structure.

### 4.3. Electronic Structure and Bonding

The charge density differences reflected the bonding characteristics through the electric charge transference, which was a critical analysis method for the interface bonding. Therefore, to gain further insight into the bonding feature of the interface, the charge density differences of the four HCP stacking structures after full relaxation are shown in Figure 8. From Figure 8, the charges were distributed more intensively at the interface because of the interfacial charge redistribution and the localised characteristics of the charge transfer. However, the lost charge of the interior atoms distributed around the atoms regularly and presented a slight distortion because of the atomic interaction. Moreover, although chemical bonds were formed among the interfacial Mg, Si, and Sb atoms, the bond strength was different. 

For the Mg–HCP–Mg interface, as shown in Figure 8a, a lower charge density was distributed between the interfacial Mg atom of the Mg_2_Si side and the Mg atom of the Mg_3_Sb_2_ side; this led to the certain ionic feature on the interface. For the Mg–HCP–Sb interface, as shown in Figure 8b, the stronger bonding strength at the interface was obviously observed, and the charge depletion mainly existed near the interfacial Mg atom of the Mg_2_Si side and the Sb atom of the Mg_3_Sb_2_ side, which indicates that the ionic bonding is formed between the interfacial Mg atom and Sb atom. For the Si–HCP–Mg interface, as shown in Figure 8c, the charge accumulation between the interfacial Si atom and the Mg atom of the Mg_3_Sb_2_ side was observed, which implies that covalent and covalent bonding in the Si–HCP–Mg interface may have existed. Figure 8d shows that large charges were accumulated in the Si–HCP–Sb interface and the strongest bonding strength, which proves that the metallic and covalent bonds may have formed at the interface. This resulted in a stronger bonding strength at the interface and explained well why among all the interface models, the Si–HCP–Sb interface had the smallest d_0_ and the highest W_ad_ values. All of these results were highly consistent with the work of adhesion and the interfacial energy, as mentioned in the previous section.

In order to have a further insight into the electronic structure and the interfacial bonding mechanism of the Mg_3_Sb_2_(0001) and Mg_2_Si(111) interface, the partial density of states (PDOS) of four different HCP interface structures was investigated, as shown in Figure 9. 

In Figure 9a, for the Mg–HCP–Mg interface, the interfacial Mg atoms had an obvious non-localised feature, which indicated that the Mg–HCP–Mg interface had stronger metallic features. However, higher DOS values of the interfacial Mg atom of Mg_2_Si and Mg_3_Sb_2_ at the Fermi level signified the presence of electron hybridisation at the interface, which resulted in a lower bonding strength of the Mg–HCP–Mg interface. A comparison of the PDOS of the interfacial Mg atom, Sb atom, and Si atom in the different layers in Figure 9b revealed that the PDOS curves of the interfacial Mg atom of the Mg_2_Si side and the Sb atom of the Mg_3_Sb_2_ side were obviously different from those of the interior layers. An obvious orbital hybridisation was observed between the interfacial Sb-s and Si-s states in the two obvious peaks at −7.35 eV and −9.10 eV, respectively, which indicated that the covalent bond was formed at the interface. Simultaneously, the DOS values for the interfacial Mg atom and the Sb atom of Mg_3_Sb_2_ increased by varying degrees, and those for the interfacial Mg atom and the Si atom of Mg_2_Si decreased, which led to the appearance of a metallic feature in the interface bonding. Therefore, metallic bonds and covalent bonds coexisted in the Mg–HCP–Sb interface. In Figure 9c, for the curves of the Sb-p orbitals, the interfacial Sb atom on the Mg_3_Sb_2_ side had more occupied states than the interior Sb atoms near the interface, which indicated that the interfacial Sb atom had significant metallic bonding at the Si–HCP–Mg interface. In addition, the covalent bond was formed because of the hybridisation between the interfacial Sb-sp state, Si-s state, and Mg-p state in the range of −8.17 to −6.47 eV, −9.76 to −8.52 eV, and −8.26 to −6.52 eV, respectively. Therefore, mixed covalent and metallic bonds also existed at the Si–HCP–Mg interface. For the Si–HCP–Sb interface, as shown in Figure 9d, a comparison of the PDOS curves of the interfacial Si atom and the Sb atom revealed the strong hybridisation between the interfacial Sb-sp and Si-sp orbits. Moreover, the PDOS curves of the interfacial Si atoms were similar to those of the interfacial Sb atoms from −12.0 eV to −2.1 eV. All this indicated that the Si–HCP–Sb interface had strong covalent bonding, which elucidated well the stronger covalent bonding resulting in a higher W_ad_.

### 4.4. Heterogeneous Nucleation Analysis of Mg_3_Sb_2_/Mg_2_Si

According to the above-calculated result, the Si–HCP–Sb interface was the most stable interface to be the heterogeneous nucleus of Mg_2_Si among all the 12 interface models, because of its smallest interfacial energy and highest work of adhesion. Although the calculated properties of the Mg_3_Sb_2_(0001) and Mg_2_Si(111) interface, such as adhesion work and interfacial energy, were all obtained at 0K, the calculated results were verified to be accurate and practically acceptable for the solid–solid and solid–liquid interfaces at high temperatures [48]. Therefore, the present calculated results theoretically validated the experimental [23] conclusion at high temperatures for the heterogeneous nucleation of Mg_3_Sb_2_ on Mg_2_Si in Mg–Si alloys.

## 5. Conclusions

To reveal the mechanism of the Mg_3_Sb_2_ heterogeneous nucleation on Mg_2_Si in Mg–Si alloys, the properties of bulk, interface stability (adhesion energy and interfacial energy), and electronic structure and bonding of Mg_3_Sb_2_(0001) /Mg_2_Si(111) were calculated by using the first-principles methods. Four types of terminations and three interfacial atom stacking sites were compared to investigate the heterogeneous nucleation efficiency of Mg_3_Sb_2_ (0001) on Mg_2_Si(111). The main conclusions were as follows:
(1)For both the Mg_2_Si (111) slab and the Mg_3_Sb_2_ (0001) slab, the 11-layered surface achieved bulk-like characteristics. The Sb-terminated Mg_3_Sb_2_ (0001) surface and the Si-terminated Mg_2_Si (111) surface were more stable than the Mg-terminated surface because of the lower surface energy.(2)Compared with all the stacking sequences, the hollow-stacked interfaces were the most stable interface. Moreover, compared with all the terminated interfaces, the Si–HCP–Sb interface was the most stable interface, because of the fact that W_ad_ and the interface spacing of the Si–HCP–Sb interface, Si–HCP–Mg interface, Mg–HCP–Sb interface, and Mg–HCP–Mg interface were 2.54 J/m^2^ and 0.9 Å, 2.05 J/m^2^ and 1.6 Å, 1.51 J/m^2^ and 1.5Å, and 0.86 J/m^2^ and 1.3Å, respectively.(3)The chemical bonding of the Mg–HCP–Mg interfaces presented stronger metallic bonding, which exhibited the highest interfacial energy. The Mg–HCP–Sb interface and the Si–HCP–Mg interface bonding similarly exhibited a mixture of covalent and metallic bonds. In particular, the Si–HCP–Sb interfaces had an obvious strong covalent feature and the smallest interfacial energy, which showed the largest stability interface among the 12 interface models.

## Figures and Tables

**Figure 1 materials-13-01681-f001:**
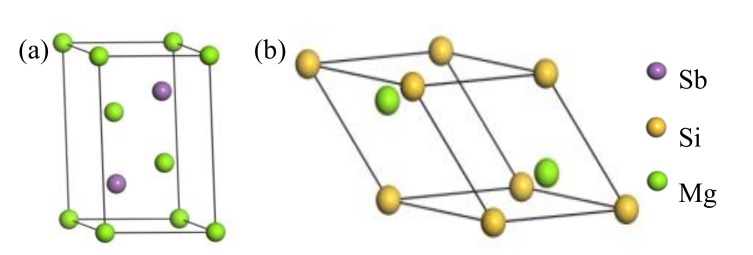
The crystal structure of (**a**) Mg_3_Sb_2_ and (**b**) Mg_2_Si.

**Figure 2 materials-13-01681-f002:**
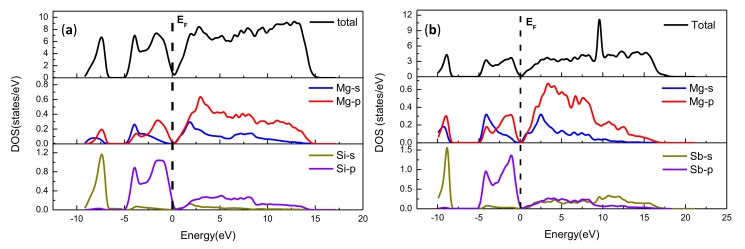
Partial density of state (DOS) charts of (**a**) Mg_2_Si and (**b**) Mg_3_Sb_2_.

**Figure 3 materials-13-01681-f003:**
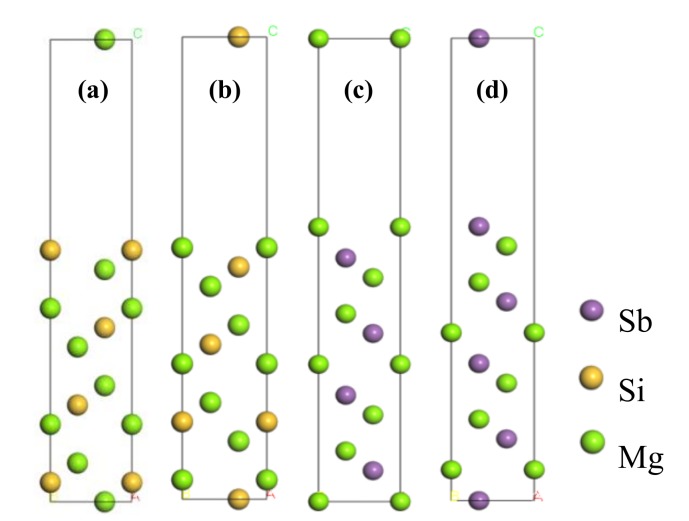
Four-surface model of (**a**) Si-terminated Mg_2_Si (111) slab, (**b**) Mg-terminated Mg_2_Si (111) slab, (**c**) Mg-terminated and Mg_3_Sb_2_ (0001) slab, and (**d**) Sb-terminated Mg_3_Sb_2_ (0001) slab.

**Figure 4 materials-13-01681-f004:**
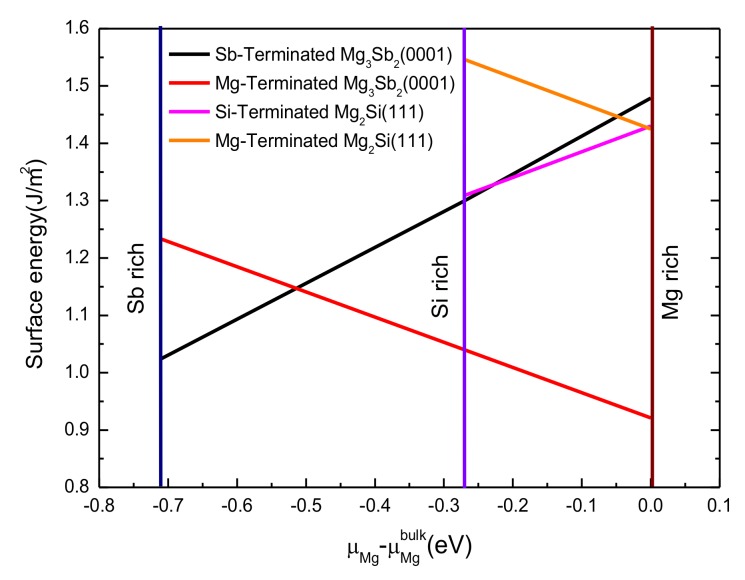
Calculated surface energy of Mg_2_Si (111) and Mg_3_Sb_2_ (0001) as a function of the magnesium chemical potential.

**Figure 5 materials-13-01681-f005:**
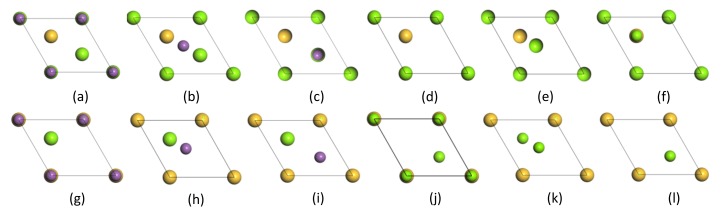
Top views of 12 different Mg_2_Si (111)/Mg_3_Sb_2_ (0001) interface models: (**a**–**c**) Top, Bridge, and Hollow sites of Mg-terminated Mg_2_Si (111) and Sb-terminated Mg_3_Sb_2_ (0001), (**d**–**f**) Top, Bridge, and Hollow sites of Mg-terminated Mg_2_Si (111) and Mg-terminated Mg_3_Sb_2_ (0001), (**g**–**i**) Top, Bridge, and Hollow sites of Si-terminated Mg_2_Si (111) and Sb-terminated Mg_3_Sb_2_ (0001), and (**j**–**l**) Top, Bridge, and Hollow sites of Si-terminated Mg_2_Si (111) and Mg-terminated Mg_3_Sb_2_ (0001).

**Figure 6 materials-13-01681-f006:**
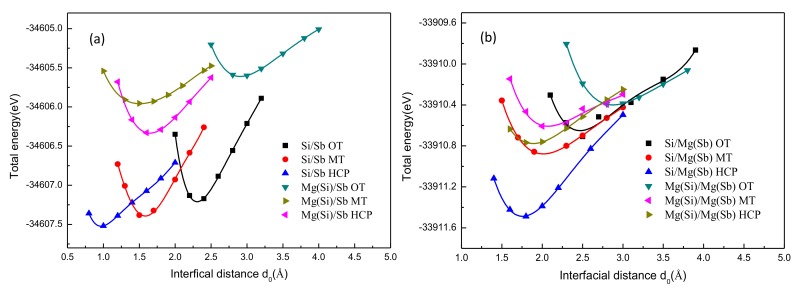
Total energy of the interfacial supercell as a function of the interface distance for 12 different models: (**a**) Sb-terminated Mg_3_Sb_2_ (0001) models and (**b**) Mg-terminated Mg_3_Sb_2_ (0001) models.

**Figure 7 materials-13-01681-f007:**
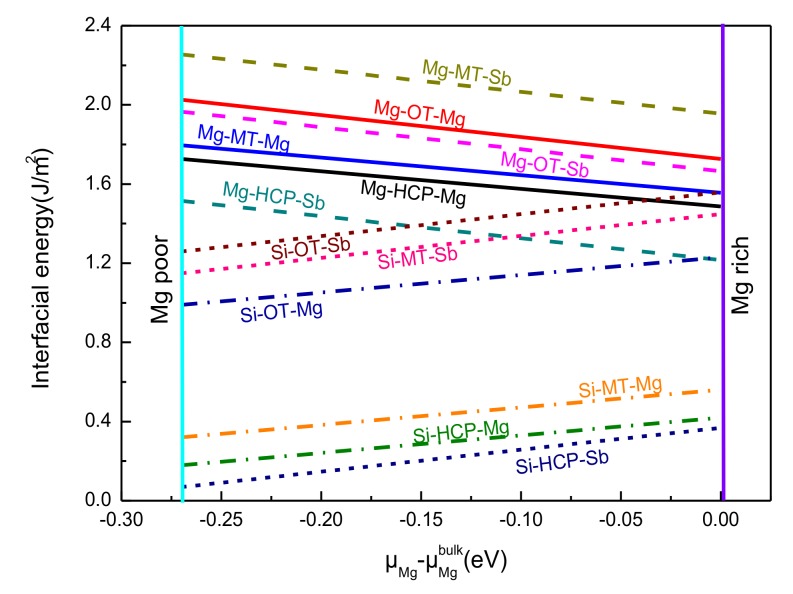
Interfacial energies of 12 interface systems as a function of the magnesium chemical potential.

**Figure 8 materials-13-01681-f008:**
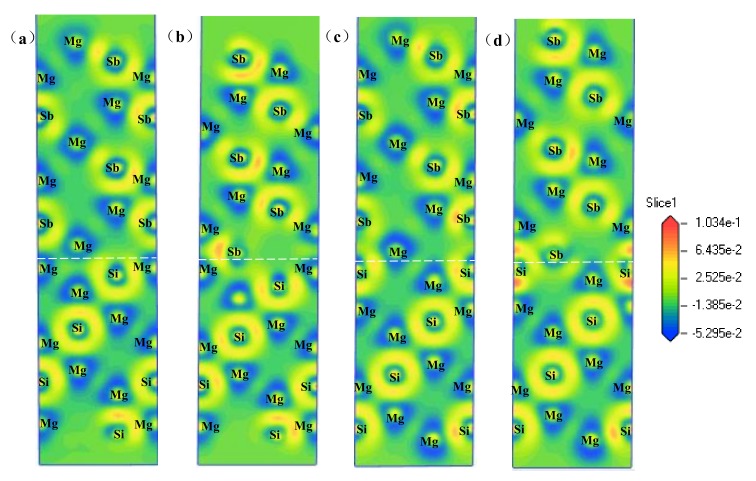
Charge density differences (e/A^3^) after full relaxation for the (**a**) Mg-HCP-Mg interface, (**b**) Mg-HCP-Sb interface, (**c**) Si-HCP-Mg interface, and (**d**) Si-HCP-Sb interface.

**Figure 9 materials-13-01681-f009:**
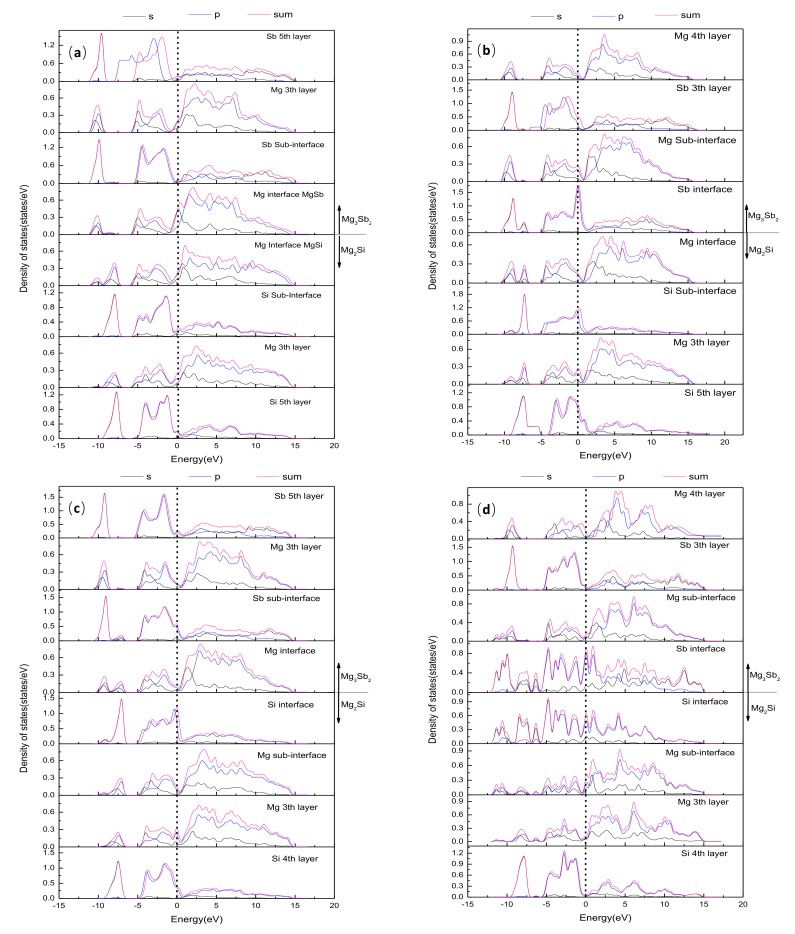
The layer-projected partial density of states (PDOS) for the Mg_3_Sb_2_(0001)/Mg_2_Si(111) interface with hollow-sited stacking. The (**a**) Mg–HCP–Mg interface, (**b**) Mg–HCP–Sb interface, (**c**) Si–HCP–Mg interface, and (**d**) Si–HCP–Sb interface. The dotted line refers to the Fermi level.

**Table 1 materials-13-01681-t001:** Calculated and experimental value of the lattice constants, bulk modulus and formation energy of Mg_2_Si and Mg_3_Sb_2_.

System	Method	Space Group	Elastic Constants					Lattice Constants		Bulk Modulus	Formation Energy
			C_11_	C_12_	C_13_	C_44_	C_66_	α/Å	c/Å	B_0_/GPa	E_for_ (eV)
Mg_2_Si	This work	Fm$\bar{3}$m	13.4	25.3		47.9		6.365	6.365	54.3	−2.24
	Other works		11.6	23.7		49.5^34^		6.346	6.346	55.3^3^^4^	−2.39
	Experiment		13.2	26.3		48.5^35^		6.350	6.350	57.3^3^^5^	
Mg_3_Sb_2_	This work	P$\bar{3}$m1	41.5	86.7	48.5	16.1	18.9	4.592	7.272	43.1	−2.12
	Other works		40.4^36^	84.4^36^	46.7^36^	15.4^36^	17.6^36^	4.573^3^^6^	7.229^3^^6^	43.9^3^^6^	−2.54
	Experiment							4.606^3^^7^	7.295^3^^7^	45.3^3^^7^	

**Table 2 materials-13-01681-t002:** The interlayer relaxation change (Δ_ij_) convergence of Mg_2_Si (111) and Mg_3_Sb_2_ (0001) with respect to the termination and atom layers.

Surface	Termination	Interlayer	Slab Thickness, N			
			5	7	9	11
Mg_2_Si(111)	Mg	Δ_12_	−13.2	−12.35	8.79	−8.047
		Δ_23_	4.53	7.98	−7.96	7.31
		Δ_34_		−1.99	−4.68	−1.15
		Δ_45_			0.72	1.43
		Δ_56_				0.048
	Si	Δ_12_	−15.02	−16.24	−15.69	−9.1
		Δ_23_	7.45	12	8.46	3.65
		Δ_34_		0.89	3.4185	3.13
		Δ_45_			−1.01	−1.14
		Δ_56_				0.62
Mg_3_Sb_2_(0001)	Mg	Δ_12_	−13.52	−12.55	−16.3	−11.8
		Δ_23_	11.24	11.85	9.22	−8.62
		Δ_34_		8.66	−6.23	1.65
		Δ_45_			2.00	−0.65
		Δ_56_				−0.32
	Si	Δ_12_	−12.56	10.63	11.68	16.92
		Δ_23_	7.31	−6.31	−5.33	10.96
		Δ_34_		−0.57	−4.126	4.43
		Δ_45_			−1.23	−1.86
		Δ_56_				−0.51

**Table 3 materials-13-01681-t003:** The Interfacial distance (*d*_0_) and interfacial energy (*y*_int_) after full relaxation.

Termination		Stacking	Fully Relaxed	
Mg_2_Si (111)	Mg_3_Sb_2_ (0001)		d_0_/Å	W_ad_(J/m^2^)
Mg-Terminated	Mg-Terminated	OT	2.6	0.56
		MT	1.8	0.79
		HCP	1.3	0.86
Mg-Terminated	Sb-Terminated	OT	2.6	0.77
		MT	1.4	1.06
		HCP	1.5	1.51
Si-Terminated	Mg-Terminated	OT	2.4	1.24
		MT	1.8	1.91
		HCP	1.6	2.05
Si-Terminated	Sb-Terminated	OT	2.2	1.35
		MT	1.3	1.46
		HCP	0.9	2.54

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
