# Peer review of "Investigation on Mg3Sb2/Mg2Si Heterogeneous Nucleation Interface Using Density Functional Theory"

_materials, 2020, doi:10.3390/ma13071681_

Round 1
Reviewer 1 Report
Wang and coworkers have explored the interfacial properties between Mg2Si(111) and Mg3Sb2(0001) using density functional theory. This is of interest for technologically important Mg-Al alloys. The authors have already explored the Mg2Si/Mg3Sb2 interfaces (Ref. 24), but it is not clear if the same orientation was considered or not. If the same orientation is considered in the current manuscript, it should directly be rejected. However, if the orientations are not the same, the authors should clearly specify it in the introduction and provide additional motivation for the new (additional) interface explored in the current work. In the latter case, there are more issues to address, which are listed below.
1) Please provide a full crystallographic description of Mg2Si and Mg3Sb2 (space groups and prototypes).
2) The electronic structure of Sb is 4d10 5s2 5p3. However, the authors have written that its electronic structure is 5s2 5p3 4d5, which is wrong. This would imply a half-filled d shell of Sb and probably important spin polarization effects. This is unphysical and raises serious questions about the manuscript. Along the same lines, the d states of Sb should have no influence on the chemical bonding and must be removed from Fig. 1 and Fig. 8.
3) Are PW91 projector augmented waves and not simple pseudopotentials? The authors have claimed that these are ultrasoft pseudopotentials. Please clarify it.
4) The formation energy in Table 1 is positive indicating that Mg2Si and Mg3Sb2 are unstable. This does not seem sound. Furthermore, how are formation energy data in Table 1 related to Eq. (5)? Also, there are no elastic constants in Table 1.
5) Are there any lateral relaxation effects? The interfaces in Fig. 2 seem to be too thin in the x-y plane.
6) Please use the Greek symbols in Eq. (3), Eq. (4), and Eq. (7).
7) The designations “(a)” and “(b)” in Fig. 1 are not properly aligned.
8) Please compare the calculated surface energy with the literature, at least with pure Mg, Al, etc.
9) Please provide the surface energy as a function of number of surface layers in Table 2. This is important for the convergence issues.
10) Please clearly define OT, MT, and HCP.
11) Fig. 5 is redundant as it contains the unrelaxed data.
12) There are fundamental errors regarding the electronic structure analysis. Charge transfer is related to ionic bonding and not to covalent or metallic bonding (“charge transfer occurred on the interface and formed an Mg-Mg metallic bond” and “charge transfer from interfacial Si atom to the Mg atom of Mg3Sb2 side is obviously observed, which implies the strong covalent bonding exists in the Si - HCP - Mg interface”). Furthermore, peaks at the Fermi level may lead to electronic instabilities rather than to stabilization (see e.g. Ravindran and Asokamani in Bull. Mater. Sci. 20, 613 (1997)).
13) The discussion on coherency and misfit (section 4.4.) should be moved up.
14) Units are missing in Fig. 7.
15) English is very poor. The authors tent to make up their own terms (“nucleation potency”, “stacking interface”, “atomic transference”, …), mix British and American English (“aluminium” vs. “aluminum”), use vague formulations (“grain improvement”, “experimental expectations”, …), some sentences are not even finished (“In other words, the initially three stacking sequences interfacial structure are”), there are many typos (“tow different termination structures”), there is often no difference between verbs, nouns, and adjectives (“various terminated of atoms”), and many sentences are not understandable due to errors (“Generally, the surface slab is equilibrium with bulk structure after fully relaxed, so the chemical potentials of Mg2Si (1 1 1) plane can be expressed by bulk Mg2Si, which is equated as[40]:”).
Reviewer 2 Report
See the attached file

Reviewer 3 Report
The manuscript is an interesting piece of work and could be publishable following some minor amendments. Further development of the software used and the experimental hardware, error analysis, and the nomenclature have to be re-examined (e.g. eq. (2), (3), (4), etc). The findings may be discussed further in the conclusions, the limitations of the research and future proposals.
Round 2
Reviewer 1 Report
The authors have improved the manuscript to some extent, but there are still open issues.
1) The authors have not supplied the full crystallographic information about the explored phases (space group, prototype). A figure doesn’t help too much. Without such information, a reader cannot know if interfaces are properly described or how many independent elastic constants are expected (according to Table 1, Mg2Si has only two, which is impossible – the highest symmetry crystals (cubic) have three: C11, C12, and C44).
2) Please remove the Sb d states from Fig. 2 and Fig. 9.
3) Some conclusions are still wrong. Besides grammatical errors, the following statement is wrong “…the charge transfer from the interfacial Si atom to the Mg atom of the Mg3Sb2 side was obviously observed, which implied that there may existed covalent bonding in the Si–HCP–Mg interface.” A charge transfer implies ionic bonding and not covalent.
4) The electronic instabilities (peaks at the Fermi level) in Fig. 9 are ignored. They are probably related to antibonding.
5) English is still poor (“stacking interface”, “atomic transference”, ???).
Reviewer 2 Report
Please find attached my report. The manuscript needs some more work before it can be accepted for publication.

Round 3
Reviewer 1 Report
The manuscript is acceptable in the current form.